# KrwEmd: Revising the Imperfect Recall Abstraction from Forgetting Everything

## Abstract

Excessive abstraction is a critical challenge in solving games with ordered signals—a subset of imperfect information games—that impairs AI performance. This issue is caused by extreme implementations of imperfect recall, which discard historical information. This paper presents KrwEmd, the first practical algorithm to address this issue. We first introduce the k-recall winrate feature, which not only qualitatively distinguishes signal infosets by leveraging future and, more importantly, historical game information, but also quantitatively reflects their similarity. We then develop the KrwEmd algorithm, which clusters signal infosets using Earth Mover's Distance to assess discrepancies between their features. Experimental results show that KrwEmd significantly enhances AI gameplay performance compared to existing algorithms.

## 1 Introduction

Abstraction refers to the process of simplifying complex games by grouping similar states or actions into broader categories, thereby improving decision-making and computational efficiency. Among these methods, imperfect recall abstraction further enhances computational efficiency by relaxing the memory consistency constraint on solvers. Recently, artificial intelligence systems employing imperfect recall abstraction have successfully developed strategies that outperformed human experts in no-limit Texas Hold'em poker, a popular testbed for imperfect information games, even under limited computational resources (Moravčík et al., 2017; Brown & Sandholm, 2018; 2019).

The hand abstraction task in Texas Hold'em can be framed as an unsupervised representation learning process, where we aim to learn efficient representations of game states by grouping similar hands. These representations enable the AI to generalize across different but related scenarios, applying a unified strategy to abstracted groups of hands, ultimately simplifying the game-solving computations. In imperfect recall setting (Waugh et al., 2009b; Johanson et al., 2013), hand abstraction in the late game does not strictly depend on the results of hand abstraction in the early game. Due to considerations of computational simplicity, imperfect recall abstraction is frequently implemented in an extreme manner, completely disregarding past memory and focusing solely on future information (Gilpin & Sandholm, 2006; 2007a; Gilpin et al., 2007; Gilpin & Sandholm, 2008; Ganzfried & Sandholm, 2014). Although these implementations reduce computational complexity, the loss of historical information hin-

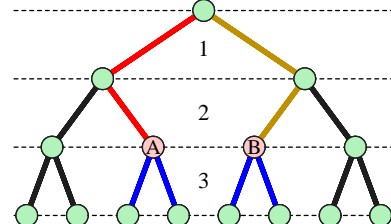

Figure 1: In a 3-phase game hand abstraction task, the current goal is to group hands A and B, which share the same future trajectory despite following different historical paths. When using a future considered only approach, both hands are assigned identical features.

ders solvers' performance by limiting the AIs' ability to maintain a comprehensive global perspective. Recent research (Fu et al., 2024) has shown that constructing hand features, used to categorize hands, solely based on future information—often referred to as **future considered only**—can lead to **excessive abstraction**, where hands with significant differences are often grouped into the same category, as shown in Figure 1. As the game progresses, the issue becomes increasingly apparent, leading to a spindle-shaped distribution of distinct features: fewer in the early and late phases, with a peak in the middle phases. This pattern fails to capture the continuous rise in the number of equivalence classes of hands throughout the game. Constructing hand features with historical information

in addition to future data can mitigate excessive abstraction by enriching the historical details of features and allowing for finer distinctions between hands.

However, two unresolved issues remain. First, as introduced by Fu et al. (2024), the k-recall outcome feature incorporates historical information. While it can be determined whether two features are identical, it lacks the ability to discern the extent of differences between features. Consequently, the k-recall outcome isomorphism (KROI) identified using this feature cannot be further refined using clustering algorithms, such as k-means, to adjust the number of categories, making it challenging to develop an effective hand abstraction algorithm that integrates historical information. Second, due to the inability to adjust the number of centroids, Fu et al. (2024) only compared the performance between two maximum centroid cases: one with the integration of historical information (KROI) and the other without (potential outcome isomorphism, POI). Although KROI significantly outperforms POI in this scenario, the comparison is inconclusive because KROI identifies more abstracted infosets than POI. Therefore, it does not conclusively prove that abstraction algorithms integrating historical information are necessarily superior when the number of abstracted infosets is the same.

This paper presents a framework for constructing hand features based on winrates, particularly the k-recall winrate feature, which utilizes significantly fewer data but still achieves approximately 90% of the resolution of KROI through its derived k-recall winrate isomorphism. By combining the earth mover's distance with the k-recall winrate feature, we developed KrwEmd, the first hand abstraction algorithm that integrates historical information, and designed an efficient computational method. We validated our approach in the Numeral211 game environment, where KrwEmd demonstrated superior performance to POI under the same abstracted infosets number condition. Additionally, in clustering scenarios, KrwEmd also outperformed other imperfect recall abstraction algorithms.

## 2 BACKGROUND AND NOTATION

Generally, Texas Hold'em-style poker games are modeled as imperfect-information games. However, for the task of hand abstraction, games with ordered signals (Gilpin & Sandholm, 2007b; Fu et al., 2024) offer a better theoretical tool. The game with ordered signals is a subclass of imperfect-information games, where the nodes (also referred to as histories, states, or trajectories) are further subdivided into two mutually independent parts: signals and public nodes. This allows for each aspect to be studied in isolation. Under this framework, the hand abstraction task in Texas Hold'em-style games is modeled as signal abstraction. Heads-up limit Texas Hold'em (HULHE) and heads-up no-limit Texas Hold'em (HUNL) are important AI testbeds. The rules for HULHE are provided in the Appendix A, and the dealing rules for HUNL are the same as those in HULHE.

### 2.1 GAMES WITH ORDERED SIGNALS

**Definition 1.** A structure $\tilde{\mathcal{G}} = \left\langle \tilde{\mathcal{T}}, \tilde{N}, \rho, \tilde{A}, Pa_{\tilde{A}}, \gamma, \Theta, \varsigma, \vartheta, \omega, \succeq, u \right\rangle$ formally defines a **game with ordered signals**, where:

- $\tilde{\mathcal{T}} = \left\langle \tilde{V}, \tilde{v}^0, \tilde{Z}, Pa_{\tilde{V}} \right\rangle$ is a public tree consisting of a finite set of public nodes $\tilde{V}$, a unique initial node $\tilde{v}^0 \in \tilde{V}$, a finite set of terminal public nodes $\tilde{Z} \subseteq \tilde{V}$, and a predecessor function $Pa_{\tilde{V}} : \tilde{V}_+ \to \tilde{H}$, mapping each non-initial node $\tilde{v}_+ \in \tilde{V}_+$ to its immediate predecessor $\tilde{h} \in \tilde{H}$. Here, $\tilde{V}_+ = \tilde{V} \backslash \{\tilde{v}^0\}$ is the set of non-initial nodes, while $\tilde{H} = \tilde{V} \backslash \tilde{Z}$ is the set of internal nodes.

- $\tilde{N} = N \cup \{sp\}$ is a finite set of augmented players, where $sp$ refers to a spectator who observes the public information of the game without influencing its progression. $N = \{0, 1, \ldots, n\}$ denotes the set of players, with $0$ representing a special player, commonly referred to as **chance** or **nature**, whose actions correspond to random events. The set of rational players (i.e., non-chance players) is denoted by $N_+ = N \backslash \{0\}$, and the set of augmented rational players is given by $\tilde{N}_+ = N_+ \cup \{sp\}$. The player function $\tilde{\rho} : \tilde{H} \to N$ partitions the set $\tilde{H}$ among players. The set of decision public nodes is defined as $\tilde{H}_+ = \bigcup_{i \in N_+} \tilde{H}_i$, where $\tilde{H}_i = \{\tilde{h} \in \tilde{H} \mid \rho(\tilde{h}) = i\}$, while the set of chance public nodes is given by $\tilde{H}_0 = \tilde{H} \backslash \tilde{H}_+$.

- $\tilde{A} = A_+ \cup \tilde{A}_0$ is a finite set of actions. The set $A_+$ includes the actions available to rational players, while $\tilde{A}_0 = \{a_0\}$ denotes the set of actions available to the chance player. Notably, $\tilde{A}_0$ contains

only one action, $a_0$, which represents a placeholder action in the public tree where the chance player reveals a signal. The function $Pa_{\tilde{A}} : \tilde{V}_+ \to \tilde{A}$ defines an action partition of $\tilde{V}_+$, mapping each non-initial public node $\tilde{v}_+$ to the action $a \in \tilde{A}$ that immediately leads to the occurrence of $\tilde{v}_+$. The function $\mathcal{A}(\tilde{h} \in \tilde{H}) = \{a' \in \tilde{A} \mid [\exists \tilde{v}_+ \in \tilde{V}_+](Pa_{\tilde{A}}(\tilde{v}_+) = a' \wedge Pa_{\tilde{V}}(\tilde{v}_+) = \tilde{h})\}$ confines the available actions of each internal public node.

- $\gamma : \tilde{V} \to \mathbb{N}^+$ is a phase partition of $\tilde{V}$, assigning to each public node $\tilde{v}$ a value corresponding to the number of chance public nodes encountered along the path from the initial public node $\tilde{v}^0$ to and including $\tilde{v}$, thereby defining the phase of $\tilde{v}$. The maximum phase in the game is denoted by $\Gamma$, and notably, $\gamma(\tilde{v}^0) = 1$, indicating that $\tilde{v}^0$ is a chance public node.

- $\Theta = \langle \Theta, \theta^0, Pa_\Theta \rangle$ is a signal tree with height $\Gamma$, consisting of a finite set of signals $\Theta$, a unique initial signal $\theta^0$, and a predecessor function $Pa_\Theta : \Theta^{(+)} \to \Theta \setminus \Theta^{(\Gamma)}$, mapping each non-initial signal to its immediate predecessor. Here, $\Theta^{(r)}$ denotes the set of signals revealed in phase $r = 1, \ldots, \Gamma$; specifically, $\Theta^{(0)} = \{\theta^0\}$. $\Theta^{(+)} = \Theta \setminus \Theta^{(0)}$ represents the set of non-initial signals. The depth of a signal $\theta \in \Theta$ is denoted by $d_\Theta(\theta)$, and all terminal signals in $\Theta$ (i.e., signals without successors) necessarily have a depth of $\Gamma$.

- $\varsigma : \Omega \mapsto [0, 1]$ is a chance probability function that assigns a probability of occurrence to each successive pair of signals, with $\Omega = \{(\theta, \theta') \in \Theta \times (\Theta \setminus \Theta^{(\Gamma)}) \mid Pa_\Theta(\theta') = \theta\}$. Additionally, for each $\theta \in \Theta \setminus \Theta^{(\Gamma)}$, the sum of the probabilities for all $\theta' \in S(\theta)$ is equal to 1, where $S(\theta) = \{\theta' \in \Theta^{(+)} \mid Pa_\Theta(\theta') = \theta\}$.

- $\vartheta = (\vartheta_1, \ldots, \vartheta_n, \vartheta_{sp})$ is a tuple of observation functions, with $\vartheta_i : \Theta \mapsto \Psi_i$ mapping each $\theta \in \Theta$ to its corresponding signal infoset (i.e., information set), such that signals within the same signal infoset $\psi \in \Psi_i$ cannot be distinguished by the augmented rational player $i \in \tilde{N}_+$. Furthermore, all elements $\psi \in \Psi_i$ collectively form a partition of $\Theta$.

- $\omega = (\omega_1, \ldots, \omega_n)$ is a tuple of survival functions, where $\omega_i(\tilde{v} \in \tilde{V}) := \mathbf{1}_{\text{player } i \text{ is still participating at } \tilde{v}}$.

- $\succeq$ is a total order over the terminal signals with respect to the set of players $N_+$, where $\succeq (\theta \in \Theta^{(r)}, i \in N_+, j \in N_+) := \mathbf{1}_{\text{player } i \text{ is ranked no lower than player } j \text{ at } \theta}$.

- Signals and public nodes constitute the nodes of an ordered game. The corresponding sets are defined as follows:
  - $H_0^{(r)} = \tilde{H}_0^{(r)} \times \Theta^{(r-1)}$, and for $j \in N_+$, $H_j^{(r)} = \tilde{H}_j^{(r)} \times \Theta^{(r)}$, $r = 1, \ldots, \Gamma$, where $\tilde{H}_i^{(r)} = \{\tilde{h} \in \tilde{H}_i \mid \gamma(\tilde{h}) = r\}$, $i \in N$.
  - $Z^{(r)} = \tilde{Z}^{(r)} \times \Theta^{(r)}$, where $\tilde{Z}^{(r)} = \{\tilde{z} \in \tilde{Z} \mid \gamma(\tilde{h}) = r\}$, $r = 1, \ldots, \Gamma$.
  - $H^{(r)} = \bigcup_{i \in N} H_i^{(r)}$ and $V^{(r)} = Z^{(r)} \cup H^{(r)}$.
  - $H = \bigcup_{r=1}^\Gamma H^{(r)}$, $Z = \bigcup_{r=1}^\Gamma Z^{(r)}$, and $V = \bigcup_{r=1}^\Gamma V^{(r)}$.

- $u = (u_1, \ldots, u_n)$ is a tuple of utility functions, where $u_i : Z \mapsto \mathbb{R}$. In the final phase, for $z = (\tilde{z}, \theta) \in Z^{(\Gamma)}$, it is required that if $\omega_i(\tilde{z})\omega_j(\tilde{z}) \succeq (\theta, i, j) = 1$, then $u_i(\tilde{z}, \theta) \geq u_j(\tilde{z}, \theta)$.

## 2.2 STRATEGIES AND NASH EQUILIBRIUM IN GAMES WITH ORDERED SIGNALS

Rational players make decisions based on their observations of signals (i.e., signal infoset) and the current non-terminal public node. Signals within the same signal infoset necessarily share the same depth, where $d_\theta(\psi)$ denotes the depth of $\psi \in \Psi_i$. A rational player has access to more information than the spectator, including all information available to the spectator. For any $i \in N_+$ and $\forall \theta \in \Theta$, we have $\vartheta_i(\theta) \subseteq \vartheta_{sp}(\theta)$.

A rational player $i \in N_+$ chooses a strategy $\sigma_i : Q_i \mapsto [0, 1]$ from $\Sigma_i$, the set of available strategies for player $i$. Here, $Q_i = \{(\tilde{h}, \psi, a) \in \tilde{H}_i \times \Psi_i \times A_+ \mid \gamma(\tilde{h}) = d_\Theta(\psi) \wedge a \in \mathcal{A}(\tilde{h})\}$, and the condition $\sum_{a \in \mathcal{A}(\tilde{h})} \sigma_i(\tilde{h}, \psi, a) = 1$ must be satisfied. When all rational players select their strategies, a strategy profile $\sigma : Q \mapsto [0, 1]$ is formed, where $\sigma = \oplus_{i \in N_+} \sigma_i \in \Sigma$ [1] and $Q = \bigcup_{i \in N_+} Q_i$, and the

---

[1] Given the functions $f_1 : A_1 \mapsto B_1$ and $f_2 : A_2 \mapsto B_2$, a new function $f = f_1 \oplus f_2$ is defined such that $f : A_1 \oplus A_2 \mapsto B_1 \cup B_2$, with

$$f(x) = \begin{cases} f_1(x) & \text{if } x \in A_1 \setminus A_2, \\ f_2(x) & \text{if } x \in A_2 \setminus A_1. \end{cases}$$

probability of reaching each node $v = (\tilde{v}, \theta) \in V$ can be compute as follow:

$$\pi(\sigma, (\tilde{v}, \theta)) = \begin{cases} 1 & \text{if } (\tilde{v}, \theta) = (\tilde{v}_0, \theta_0), \\ \varsigma(Pa_\Theta(\theta), \theta)\pi(\sigma, (Pa_{\tilde{V}}(\tilde{v}), Pa_\Theta(\theta))) & \text{if } Pa_{\tilde{V}}(\tilde{v}) \in \tilde{H}_0, \\ \sigma(Pa_{\tilde{V}}(\tilde{v}), \vartheta_i(Pa_\Theta(\theta)), Pa_{\tilde{A}}(\tilde{v}))\pi(\sigma, (Pa_{\tilde{V}}(\tilde{v}), Pa_\Theta(\theta))) & \text{if } Pa_{\tilde{V}}(\tilde{v}) \in \tilde{H}_+. \end{cases}$$

The expected payoff for rational player $i \in N_+$, given a strategy profile $\sigma \in \Sigma$, is $\hat{u}_i(\sigma) = \sum_{z \in Z} \pi(\sigma, z)u_i(z)$. A strategy profile $\sigma^* \in \Sigma$ is a **Nash equilibrium** if, for all $i \in N_+$, the following holds:

$$\hat{u}_i(\sigma^*) \geq \max_{\sigma_i' \in \Sigma_i} \hat{u}_i(\sigma^*_{-i} \oplus \sigma_i').$$

### 2.3 Signal Abstraction in Games with Ordered Signals

Abstraction is a simplified perception of the game from the player's perspective. $\alpha = (\alpha_1, ., \alpha_n)$ is a signal (infoset) abstraction profile, with $\alpha_i : \theta \mapsto \Psi_i^\alpha$, a signal (infoset) abstraction, mapping each $\theta \in \Theta$ to its corresponding abstracted signal infoset $\hat{\psi} \in \Psi_i^\alpha$ for $i \in N_+$. Each abstracted infoset $\hat{\psi}$ can be further subdivided into several signal infosets within $\Psi_i$. These finer signal infosets collectively form a partition of $\hat{\psi}$.

In general, two signal abstractions cannot be directly compared in terms of performance. However, in certain specific cases, a special relationship known as **refinement** exists between them. Consider two abstractions $\alpha_i$ and $\beta_i$. If, for every $\hat{\psi} \in \Psi_i^\beta$, there exists one or more abstracted signal infosets in $\Psi_i^\alpha$ such that their union forms a partition of $\hat{\psi}$, then $\alpha_i$ is said to refine $\beta_i$, denoted as $\alpha_i \sqsupseteq \beta_i$. The signal-abstracted game $\tilde{\mathcal{G}}^\alpha$ is derived by substituting $\vartheta_i$ with $\alpha_i$ in $\tilde{\mathcal{G}}$.

The concepts of perfect and imperfect recall are originally associated with **imperfect-information games**, indicating whether players are required to remember all the information encountered throughout the game. Since games with ordered signals are a subset of imperfect-information games, we extend the notion of signal perfect/imperfect recall to this framework. In a game $\tilde{\mathcal{G}}$, a player $i \in \tilde{N}_+$ is said to have signal perfect recall if, for any two signals $\theta_1', \theta_2' \in \psi'$, every predecessor $\theta_1$ of $\theta_1'$ corresponds to a predecessor $\theta_2$ of $\theta_2'$ such that $\theta_2 \in \vartheta_i(\theta_1)$ and $\theta_1 \in \vartheta_i(\theta_2)$. When all players in the game $\tilde{\mathcal{G}}$ have signal perfect recall, the game itself is said to have signal perfect recall. In a game with signal perfect recall, denoted by $\tilde{\mathcal{G}}$, let $\alpha_i$ represent the signal abstraction for player $i \in N_+$. The abstraction profile $(\alpha_i, \vartheta_{-i})$ refers to a scenario in which player $i$ employs the signal abstraction $\alpha_i$, while the other players do not use any signal abstraction. If the game $\tilde{\mathcal{G}}^{(\alpha_i, \vartheta_{-i})}$ retains signal perfect recall, then $\alpha_i$ is considered a signal abstraction with perfect recall; otherwise, it is considered a signal abstraction with imperfect recall.

## 3 Related Work

Our research focuses on hand abstraction techniques in AI systems for Texas Hold'em-style games (i.e. the signal abstraction in games with ordered signals), building on the foundational works of Shi & Littman (2001) and Billings et al. (2003). These seminal studies introduced game abstraction to simplify games while preserving key characteristics, initially relying on manual hand abstraction by experts. The first automated hand abstraction was developed by Gilpin & Sandholm (2006), followed by a more formal model for games with ordered signals in Texas Hold'em by Gilpin & Sandholm (2007b). They introduced the concept of lossless isomorphism (LI) through signal rotation. Despite LI's theoretical elegance, its low compression rates limited its use in large-scale games. In contrast, lossy abstractions, which balance accuracy and scalability, showed greater potential. Methods such as the expectation-based clustering (Ehs) and the histogram-based potential-aware method were later introduced by Gilpin & Sandholm (2007a) and Gilpin et al. (2007). Studies by Gilpin & Sandholm (2008) and Johanson et al. (2013) later showed that the potential-aware method outperformed Ehs in larger-scale games. In addition, Johanson et al. (2013) introduced the use of earth mover's distance (EMD) in the potential-aware method, while Ganzfried & Sandholm (2014) proposed a more efficient approximation algorithm, PaEmd, to further optimize this approach. This

methodology was further extended by Brown et al. (2015) to distributed environments, making PaEmd the state-of-the-art solution for large-scale imperfect-information games.

Recently, Fu et al. (2024) proposed several novel tools, including abstraction resolution and common refinement. They introduced two signal abstractions: potential outcome isomorphism (POI), which maximizes the number of abstracted signal infosets based on future information, and k-recall outcome isomorphism (KROI), which does so by incorporating historical information. They argued that current algorithms, which focus future considered only, tend to excessive abstraction, yet their work did not provide a practical signal abstraction algorithm, leaving an open challenge.

## 4   Winrate Isomorphism

We begin by illustrating why historical information cannot be ignored in games with ordered signals. Consider a player in HULHE with two distinct signal infosets: $[9\diamondsuit Q\heartsuit; Q\clubsuit 9\diamondsuit 9\heartsuit; K\clubsuit; 4\diamondsuit]$ and $[9\diamondsuit Q\heartsuit; K\clubsuit 4\diamondsuit 9\heartsuit; Q\clubsuit; 9\diamondsuit]$, where $[9\diamondsuit Q\heartsuit]$ is the hole cards. Despite these two signal infosets having the same hand strength, they differ in timing: in the first, the player forms a Full House by the Flop, encouraging an optimistic early-game strategy. In the second, the player completes the hand only by the River, leading to more cautious early play. A player's strategy is observed by opponents, influencing their decisions, and in turn, the opponent's decisions are observed by the player, further shaping the player's strategy. This dynamic results in a completely different game scenario and highlights that, despite the two infosets having equal strength by the River, their strategic differences prevent them from being grouped together. A player's confidence in their current signal infoset guides their strategy, and winrate is a critical factor influencing this confidence.

We introduce isomorphism frameworks for winrate-based features, including potential winrate isomorphism (PWI) and k-recall winrate isomorphism (KRWI). These frameworks serve as signal abstractions where the term **isomorphism** refers to the equivalence relations (reflexive, symmetric, and transitive) satisfied by the abstracted signal infosets. In particular, this transitivity property helps determine whether two infosets belong to the same abstracted signal infoset. To avoid ambiguity, we refer to the abstracted signal infosets defined in these frameworks (POI, KROI, PWI, KRWI) as signal infoset equivalence classes. Two infosets are classified in the same equivalence class if they share an identical defined feature.

As the name suggests, a **winrate-based feature** is a set of data that reflects the strength of a signal infoset by rolling it out to its subsequent terminal signals and comparing the players' ranks pairwise. Winrate-based features distinguish signal infosets and require far less data compared to **outcome-based features**, which use histograms representing specific outcomes as the infoset progresses to the next phase. For example, in future considered only settings in heads-up Texas Hold'em, a Preflop winrate-based feature can be represented using only three data points (win, draw, loss), whereas an outcome-based feature might require $C(50, 3)$ data points (i.e., the number of combinations of three community cards dealt from a 52-card deck after the player's two hole cards). Clearly, fewer data points reduce both time and space complexity. Nonetheless, this raises concerns about a potential loss in resolution. In this section, we argue for the use of winrate-based features and demonstrate that they do not significantly compromise resolution.

PWI and KRWI (as well as POI and KROI) share a similar isomorphism construction process, as outlined in Algorithm A1. The primary distinction between them lies in the construction operator for the (winrate-based) features, Feature, used in lines 5 and 12. The isomorphism construction process begins by iterating over all signal infosets in $\Psi_i^{(r)}$, the signal infoset space for rational player $i$ in phase $r$, and collecting their features. These features are then deduplicated and stored in lexicographical order within the set $\mathcal{C}_i^{(r)}$, implemented as a vector. In $\mathcal{C}_i^{(r)}$, the index of each feature serves as an identifier for a signal infoset equivalence class. A hash table, $\mathcal{CI}_i^{(r)}$, is then used to map a feature to its corresponding signal infoset equivalence class identifier. Finally, the algorithm revisits $\Psi_i^{(r)}$ to associate each signal infoset's identifier with that of its corresponding signal infoset equivalence class, storing this mapping in $\mathcal{D}_i^{(r)}$, the isomorphism map. The function $Index_i(r, \cdot)$ is a domain-specific mapping that assigns a unique identifier to each signal infoset at phase $r$, ranging from 0 to $|\Psi_i^{(r)}| - 1$. In Texas Hold'em-style games, one possible implementation of this function is through lossless isomorphism (Gilpin & Sandholm, 2007b; Waugh, 2013).

### 4.1 POTENTIAL WINRATE ISOMORPHISM

Potential winrate isomorphism (PWI) is a signal abstraction that classify signal infosets based on its potential winrate features. These features focus on the distribution of a player's winrate over terminal signals after passing through a given signal infoset, without considering the history of how the player reached the signal infoset. Specifically, for player $i$ in phase $r$, the potential winrate feature associated with $\psi \in \Psi_i^{(r)}$ is defined as

$$pf_i^{(r)}(\psi) = (pf_i^{(r),0}(\psi), pf_i^{(r),1}(\psi), \ldots, pf_i^{(r),n}(\psi)), \tag{1}$$

where

- $pf_i^{(r),0}(\psi)$ denotes the probability that player $i$ ranks lower than least one other player in the terminal signals, after passing through $\psi$.

- $pf_i^{(r),l}(\psi)$, for $l > 0$, denotes the probability that player $i$ ranks no lower than any other player and ranks higher than exactly $l-1$ other players in the terminal signals, after passing through $\psi$.

In the terminal phase, the winrate feature is computed by directly calculating the game outcomes for players within the given signal infoset. In contrast, during non-terminal phases, we employ a recursive approach to simplify the calculation of the winrate feature, thereby avoiding the need to enumerate every signal infoset down to the terminal phase. The recursive formula is given by

$$pf_i^{(r),l}(\psi) = \sum_{\substack{\psi' \in \Psi_i^{(r+1)} \\ \psi \sqsubseteq \psi'}} pf_i^{(r+1),l}(\psi')P(\psi|\psi'), \tag{2}$$

where $\psi \sqsubseteq \psi'$ indicates that there exist $\theta \in \psi$ and $\theta' \in \psi'$ such that $\theta$ is a predecessor of $\theta'$.

The PWI algorithm is derived from the POI algorithm (Fu et al., 2024), and the details of the PWI algorithm are elaborated in Appendix C.2. Unlike POI, PWI also uses the potential winrate feature in non-terminal phases to identify different signal infoset equivalence classes, while POI relies on the potential outcome feature (which captures the distribution of the signal infoset equivalence class for future signal infoset). In non-terminal phases, the potential winrate feature is a simplified version of the potential out-

| | Preflop | Flop | Turn | River |
|---|---|---|---|---|
| LI | 169 | 1286792 | 55190538 | 2428287420 |
| PWI | 169 | 1028325 | 1850624 | 20687 |
| POI | 169 | 1137132 | 2337912 | 20687 |
| W/O (%) | 100.0 | 90.43 | 79.16 | 100.0 |
| WD/OD | $3/C(50,3)$ | 3/47 | 3/46 | 3/3 |

Table 1: The number of signal infoset equivalence classes identified by LI, PWI, and POI in each phase of HULHE and HUNL, with W/O indicating the ratio of signal infoset equivalence classes identified by PWI to those identified by POI, and WD/OD indicating the ratio of data used by PWI to that used by POI.

come feature. Unsurprisingly, PWI also results in excessive abstraction similar to POI. As shown in Table 1, in HULHE and HUNL, the number of signal infoset equivalence classes identifiable by lossless isomorphism increases with each phase, indicating that the game becomes increasingly complex. However, the number of signal infoset equivalence classes identifiable by PWI and POI first increases and then decreases, showing a spindle-shaped pattern. And we observed that when only future information is considered, winrate-based features may lead to greater information loss compared to outcome-based features. For instance, in the River phase, the number of signal infoset equivalence classes identified by PWI is only 79.16% of that identified by POI.

### 4.2 K-RECALL WINRATE ISOMORPHISM

As Fu et al. (2024) mentioned, supplementing historical information can enhance the ability of signal abstraction to identify signal infoset equivalence classes. Inspired by KROI's construction approach, we developed the k-recall winrate isomorphism (KRWI), where k-recall refers to recalling information from the previous k phases. The key difference is that instead of using k-recall outcome features to distinguish between different signal infosets, KRWI utilizes k-recall winrate features.

In a game with signal perfect recall, all signals within the signal infoset $\psi$ have their predecessors at phase $r'$, which belong to the identical signal infoset $\psi'$. For player $i$ at phase $r$, the signal infoset

| | Preflop | Flop | | Turn | | | River | | | |
|---|---|---|---|---|---|---|---|---|---|---|
| Recall | 0 | 0 | 1 | 0 | 1 | 2 | 0 | 1 | 2 | 3 |
| KRWI | 169 | 1028325 | 1123442 | 1850624 | 34845952 | 37659309 | 20687 | 33117469 | 529890863 | 577366243 |
| KROI | 169 | 1137132 | 1241210 | 2337912 | 38938975 | 42040233 | 20687 | 39792212 | 586622784 | 638585633 |
| W/O (%) | 100.0 | 90.43 | 90.51 | 79.16 | 89.49 | 89.58 | 100.0 | 83.23 | 90.33 | 90.41 |

Table 2: The number of signal infoset equivalence classes identified by KRWI, and KROI in each phase and $k$ of HULHE and HUNL, with W/O indicating the ratio of signal infoset equivalence classes identified by KRWI to those identified by KROI.

$\psi \in \Psi_i^{(r)}$ has a $k$-recall winrate feature ($k < r$) represented as a numerical array with a dimension of $(k+1)(n+1)$:

$$rf_i^{(r,k)}(\psi) = (pf_i^{(r)}(\psi); pf_i^{(r-1)}(\psi); \ldots; pf_i^{(r-k)}(\psi)), \tag{3}$$

where $pf_i^{(r')}(\psi)$ denotes the potential winrate feature for the predecessor signal infoset $\psi'$ of $\psi$ at phase $r'$, for $r' < r$. Since we have stored all the potential winrate features of $\psi \in \Psi_i^{(r)}$ through $\mathcal{PC}_i^{(r)}, \mathcal{PD}_i^{(r)}$ and assigned them unique identifiers in Algorithm A2. To save storage space and facilitate retrieval, what we actually store is

$$rfi_i^{(r,k)}(\psi) = (\mathcal{PD}_i^{(r)}[\psi], \mathcal{PD}_i^{(r-1)}[\psi], \ldots, \mathcal{PD}_i^{(r-k)}[\psi]). \tag{4}$$

$\mathcal{PD}_i^{(r')}[\psi]$ is the identifier for the potential winrate feature of the predecessor $\psi'$ of $\psi$ in the $r'$ phase, for $r' \leq r$. For algorithm details, please refer to Appendix C.3.

Similar to how the potential winrate feature simplifies the potential outcome feature, the k-recall winrate feature is a simplified version of the k-recall outcome feature. Moreover, it is evident that 0-RWI (KRWI when $k = 0$) identifies the same infoset equivalence classes as the PWI. Table 2 presents the number of signal infosets identified by KRWI and KROI, as well as their ratio in HULHE and HUNL. Notably, while the resolution ratio of PWI to POI can fall below 80%, when $k$ is set to its maximum value, i.e., $r - 1$, the ratio of KRWI to KROI can reach nearly 90% at a minimum, with most of the information retained. Additionally, it is clear that KRWI identifies a significantly greater number of signal infoset equivalence classes than POI, which refines all signal abstraction algorithms based on the future considered only approach, such as EHS and the previous state-of-the-art PaEmd (Fu et al., 2024).

## 5 K-Recall Winrate Abstraction with Earth Mover's Distance

Building on the previously introduced winrate isomorphism framework, this section explores the application of k-recall winrate features to further abstract signal infosets. While outcome-based features focus solely on categorization, winrate-based features enable differentiation between categories by providing comparable numerical values, i.e., winrate values and vectors. Intuitively, the similarity between features corresponds to the similarity of infoset equivalence classes. Consequently, clustering algorithms can be employed to further group the infoset equivalence classes identified by PWI into appropriately sized abstracted signal infosets, facilitating their application in solving large-scale game problems.

For the signal infosets $\psi, \psi'$ of player $i$ at phase $r$, we can define the distance of their k-recall winrate feature as

$$d(rf_i^{(r,k)}(\psi), rf_i^{(r,k)}(\psi')) = \sum_{j=0}^{k} w_j \cdot \text{Emd}(pf_i^{(r-j)}(\psi), pf_i^{(r-j)}(\psi')). \tag{5}$$

Among equation 5, Emd is the operator used to calculate the earth mover's distance (EMD) (Rubner et al., 2000). The Earth Mover's Distance (EMD) can be formulated as a linear programming problem. Given two distributions $\boldsymbol{p} = (p_1, p_2, \ldots, p_n)$ and $\boldsymbol{q} = (q_1, q_2, \ldots, q_m)$ over two sets of points,

and a distane matrix $D = [d_{ij}]_{n \times m}$ representing the ground distances between each point in $\boldsymbol{p}$ and $\boldsymbol{q}$, the goal is to find the optimal flow $F = [f_{ij}]_{n \times m}$ that minimizes the total transportation cost

$$\text{Emd}(\boldsymbol{p}, \boldsymbol{q}) = \min \sum_{i=1}^{n} \sum_{j=1}^{m} w_{ij} d_{ij}$$

subject to the following constraints:

$$\sum_{j=1}^{m} f_{ij} = p_i, \quad \forall i = 1, 2, \dots, n \quad \text{(flow conservation for } \boldsymbol{p}\text{)}$$

$$\sum_{i=1}^{n} f_{ij} = q_j, \quad \forall j = 1, 2, \dots, m \quad \text{(flow conservation for } \boldsymbol{q}\text{)}$$

$$f_{ij} \geq 0, \quad \forall i, j \quad \text{(non-negativity constraint)}$$

where $f_{ij}$ represents the amount of flow from $p_i$ to $q_j$. Since it requires solving linear programming equations, the computational complexity of the EMD is sensitive to the dimensionality of the histograms, and approximate algorithms are usually used for larger-scale problems. However, the dimensionality of winrate-based features is small, with a dimension of 3 in a two-player scenario, so we attempt to use a fast algorithm for accurately computing the EMD (Bonneel et al., 2011). $w_0, \dots, w_k$ are hyperparameters used to control the importance of EMD at each phase $r, \dots, r - k$, and the idea behind this design is to transform the similarity between two infoset equivalence classes into a linear combination of the EMD distances between their k-recall winrate features' winrates across different phases. We use the KMeans++ algorithm (Arthur & Vassilvitskii, 2007) to cluster the signal infoset equivalence classes of KRWI. We named this algorithm KrwEmd.

## 6 EXPERIMENTAL SETUP

We conducted experiments on the Numeral211 Hold'em (Fu et al., 2024) testbed. Numeral211 is a two-player three-phase Taxes Hold'em-style game with more complex hand systems than the Leduc Hold'em (Southey et al., 2005) and Rhode Island Hold'em (Shi & Littman, 2001) test environments, making it suitable for studying hand abstraction issues. Detailed rules are included in Appendix B. Table 3 shows the number of signal infoset equivalence classes recognized by KRWI and KROI, along with lossless isomorphism, in Numeral211 Hold'em.

| | Preflop | Flop | | Turn | | |
|---|---|---|---|---|---|---|
| LI | 100 | 2260 | | 62020 | | |
| Recall | 0 | 0 | 1 | 0 | 1 | 2 |
| KRWI | 100 | 2234 | 2248 | 3957 | 51000 | 51070 |
| KROI | 100 | 2250 | 2260 | 3957 | 51176 | 51228 |
| W/O (%) | 100.0 | 99.29 | 99.47 | 100.0 | 99.67 | 99.69 |
| WD/OD | 3/38 | 3/37 | - | 3/3 | - | - |

Table 3: The number of signal infoset equivalence classes identified by LI, KRWI, and KROI in each phase of HULHE and HUNL, with W/O indicating the ratio of signal infoset equivalence classes identified by KRWI to those identified by KROI, and WD/OD indicating the ratio of data used by 0-RWI (PWI) to that used by 0-ROI (POI).

Let $\alpha = (\alpha_1, \alpha_2)$ be the signal abstraction we would like to assess. We will test the strength of the signal abstraction by measuring exploitability of the approximate equilibrium derived using the CSMCCFR algorithm (Zinkevich et al., 2007; Lanctot et al., 2009) in different abstracted signal infoset scales. We gauge the performance over exploitability. For doing that, we consider both symmetric and asymmetric abstraction scenarios.

In two-player games with ordered signals, **exploitability** measures the extent to which a player's strategy deviates from a Nash equilibrium. For a given strategy profile $\sigma = (\sigma_1, \sigma_2)$, the exploitability $\epsilon(\sigma)$ is computed as the difference between the game's expected total payoff at a Nash equilibrium $\sigma^*$ and the expected total payoff of the strategy being played against its best response. Formally, this is defined as

$$\epsilon(\sigma) = \frac{1}{2} \big( \max_{\sigma_1' \in \Sigma_1} \hat{u}_1(\sigma_1' \oplus \sigma_2) - \hat{u}_1(\sigma^*) + \max_{\sigma_2' \in \Sigma_2} \hat{u}_2(\sigma_1 \oplus \sigma_2') - \hat{u}_2(\sigma^*) \big),$$

which is measured in terms of milli blinds (antes) per game (mb/g) in Numeral211.

In this symmetric abstraction setting, we measure the exploitability of approximate equilibrium that is yielded when both the players in the game employ signal abstraction in the original game. However, it may lead to the abstraction pathology (Waugh et al., 2009a). To avoid such problems, we illustrate the theoretical performance of the signal abstraction under evaluation through asymmetric abstraction. The approximate equilibrium in the signal abstracted games $\tilde{\mathcal{G}}^{(\alpha_1, \vartheta_2)}$ and $\tilde{\mathcal{G}}^{(\vartheta_1, \alpha_2)}$ is obtained to obtain $\sigma^{*,1}$ and $\sigma^{*,2}$, respectively. Finally, we concatenate the two strategies to get $\sigma' = (\sigma_1^{*,1}, \sigma_2^{*,2})$ and check the exploitability of $\sigma'$.

Regarding KrwEmd, we set the distance matrix:

$$D = \begin{bmatrix} 0 & 1 & 2 \\ 1 & 0 & 1 \\ 2 & 1 & 0 \end{bmatrix}$$

For a two-player game, its meaning is quite clear. Taking the first row as an example: transitioning from a loss to a loss costs 0, transitioning to a draw costs 1, and transitioning to a win costs 2.

## 7  EXPERIMENT

Firstly, we assess the performance of KRWI (2-RWI) in comparison to other isomorphism frameworks—KROI (2-ROI), POI (0-ROI), and lossless isomorphism (LI). Note that POI is the common refinement of existing future considered only signal abstraction algorithms. Moreover, since previous works (KROI) could not control the number of abstracted infosets, they were unable to demonstrate whether incorporating historical information in signal abstraction outperformed abstraction with the same number of abstracted infosets. To address this, we included KrwEmd with the number of abstracted signal infosets set to match that of POI for a fair comparison in the isomorphism frameworks experiment. Note here, that 0-RWI and 0-ROI share the same capability of recognizing singal infoset equivalence classes in Preflop, while 0-ROI, 1-RWI, and 1-ROI show differences in identifying these equivalence classes on the Flop (with incremental improvements), but the differences are quite small, as shown in Table 3. Thus, we can directly allow clustering of KrwEmd abstraction use the signal infoset equivalence classes identified by POI in Preflop and Flop, and only perform clustering in Turn. Here, we design four sets of hyper-parameters $(w_0, w_1, w_2)$ in equation 5, i.e., exponentially decreasing: $(16, 4, 1)$, linearly decreasing: $(7, 5, 3)$, constant: $(1, 1, 1)$, and increasing: $(3, 5, 7)$ in the importance of historical information. We only show the result of best- and worst-performing parameters (to make the figure neat). The full figures appear in the Appendix E. Figure 2a shows the result of symmetric abstraction, while Figure 2b shows the result of asymmetric abstraction. We observed that although the exploitability differed between the two experiments, the relative rankings of each group remained consistent (i.e., if A outperformed B in symmetric abstraction, it also did so in asymmetric abstraction). This consistent performance across experiments indicates the absence of abstraction pathology. As expected, overfitting was observed in the symmetric abstraction scenario, though it was only significant for POI. The performance difference between 2-RWI and 2-ROI is small, which is related to the fact that the number of signal infoset equivalence classes identified by 2-RWI and 2-ROI in Numeral211 is similar (W/O generally exceeds 99%). However, in HULHE and HUNL, where W/O drops to around 90%, we believe significant differences exist. Most importantly, KrwEmd, outperforms POI—even with the worst parameter configuration(increasing importance).

Next, we compared KrwEmd's performance with the currently applied future considered only algorithms, EHS and PaEmd. It should be noted that POI is the common refinement both for Ehs and PaEmd, meaning that the maximum number of signal infoset equivalence classes they can recognize will not exceed that of POI. We set a compression rate that is 10 times lower than that of POI, while not performing abstraction for Preflop. The final number of abstracted signal infosets is set to 100, 225, 396 for Preflop, Flop and Turn. To exclude the influence of random events on performance, we generated 3 sets of abstractions for Ehs and PaEmd each. KrwEmd used hyperparameters $(w_{3,0}, w_{3,1}, w_{3,2}; w_{2,0}, w_{2,1})$ in Turn and Flop, which are exponentially decreasing $(16, 4, 1; 4, 1)$, linearly decreasing $(7, 5, 3; 5, 3)$, constant $(1, 1, 1; 1, 1)$, and increasing $(3, 5, 7; 5, 7)$ in the importance of historical information. Additionally, since PaEmd uses approximate EMD calculations, its approximate distance is asymmetric, making it difficult for the algorithm to converge. We truncated after 1000 iterations on a single core, with an average cost of 1427.7s, while Ehs and KrwEmd both

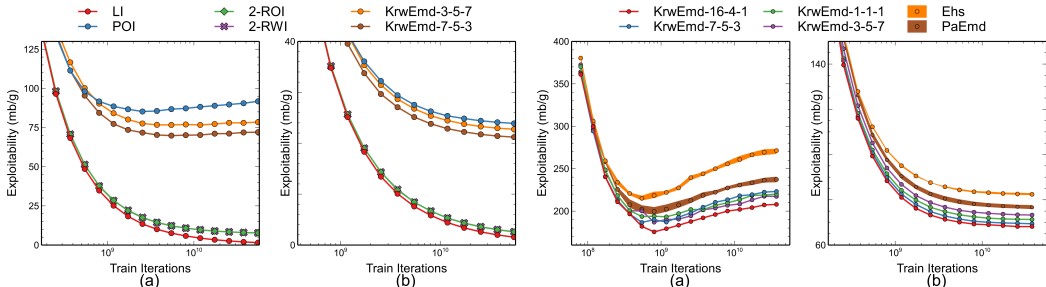

Figure 2: The isomorphism frameworks experiment was trained for $5.5 \times 10^{10}$ iterations, with (a) representing the symmetric abstraction setting and (b) representing the asymmetric abstraction setting. Both instances of KrwEmd outperform POI, while the performance of 2-RWI and 2-ROI shows almost no difference in the Numeral211 environment.

Figure 3: Performance comparison of KrwEmd versus other imperfect recall signal abstraction algorithms considering only future information, trained for $3.7 \times 10^{10}$ iterations. All instances of KrwEmd outperform the benchmark, and comparisons between KrwEmd instances indicate that late-game information is more important than early-game information.

achieved convergent clustering results, requiring an average of 12.3 and 96.7 iterations, with average time costs of 11.2s and 341.4s, respectively.

Figure 3a presents the results of the symmetric abstraction setting, while Figure 3b shows the results for the asymmetric abstraction setting. We observed that both symmetric and asymmetric abstractions maintained consistent performance, similar to the isomorphism frameworks experiment, without significant abstraction pathologies, despite noticeable overfitting in all abstraction algorithms under the symmetric setting. The experimental results indicate that KrwEmd significantly outperforms both Ehs and PaEmd across all parameter configurations. Furthermore, we validated that the importance of historical information decreases progressively from the late game to the early game, although this time the best-performing parameter decreased exponentially rather than linearly, as seen in the isomorphism frameworks experiment.

By providing a fair comparison, these two experiments validate that considering historical information is indeed more effective than the future considered only approach in signal abstraction.

## 8 CONCLUSION, LIMITATION, AND FUTURE WORK

This research introduces the first imperfect recall signal abstraction algorithm that considers historical information. This algorithm has the ability to adjust the scale of the abstracted signal infosets. Based on this, we fully verified that the imperfect recall signal abstraction algorithms considering historical information is superior to that only considering future information. Imperfect recall abstraction should be reexamined to introduce historical information and avoid excessive abstraction. Krwemd can help existing AIs achieve better performance.

KrwEmd is more competitive than previous algorithms; however, it inevitably introduces significant computational overhead. This is because KrwEmd uses the KMeans algorithm, whose time complexity scales proportionally with the size of the input data (in our case, the size of the KRWI signal infoset equivalence classes). In contrast, future considered only algorithms perform KMeans clustering using PWI as input, which is much smaller in scale. In Appendix D, we present an acceleration method that reduces the computational cost of calculating the Earth Mover's Distance in KrwEmd to a scale comparable to that of future considered only algorithms. However, the complexity of the clustering algorithm remains dependent on the size of the KRWI.

There are two potential directions for future improvements. The first is to adopt distributed computing and approximation algorithms to reduce computational complexity. The second is to explore non-KMeans algorithms and leverage machine learning techniques to incorporate historical information more effectively. Regardless of the approach, incorporating historical information in hand abstraction will help build more powerful poker game AI systems.

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

## A  HEADS-UP LIMIT TEXAS HOLD'EM RULES

| Rank | Hand | Prob. (%) | Description | Example |
|---|---|---|---|---|
| 1 | Royal flush | 0.000154 | The five highest cards (10, J, Q, K, A) of the same suit. Ties are broken by the suit. | A♠K♠Q♠J♠T♠ |
| 2 | Straight flush | 0.00139 | Five consecutive cards of the same suit. Ties are broken by the highest card. | 9♠8♠7♠6♠5♠ |
| 3 | Four of a kind | 0.0240 | Four cards of the same rank. Ties are broken by the rank of the four cards. | 9♡9♠9♢9♣K♣ |
| 4 | Full house | 0.1441 | Three cards of one rank and two of another. Ties are broken by the rank of the three cards, then the two. | Q♢Q♡Q♣9♢9♡ |
| 5 | Flush | 0.1965 | Five cards of the same suit. Ties are broken by the highest card, then the next highest, and so on. | A♡K♡7♡5♡2♡ |
| 6 | Straight | 0.3925 | Five consecutive cards, not all of the same suit. Ties are broken by the highest card. | 10♢9♡8♣7♢6♡ |
| 7 | Three of a kind | 2.1128 | Three cards of the same rank. Ties are broken by the rank of the three cards. | K♣K♡K♠J♢8♡ |
| 8 | Two pair | 4.7539 | Two cards of one rank, two of another rank. Ties are broken by the higher pair, then the lower pair. | Q♣Q♡9♠9♣5♢ |
| 9 | One pair | 42.2569 | Two cards of the same rank. Ties are broken by the rank of the pair, then the next highest card. | J♡J♠A♢7♣4♡ |
| 10 | High card | 50.1177 | None of the above. Ties are broken by the highest card, then the next highest card, and so on. | A♠K♡8♣7♢2♡ |

Table 4: Hand ranks of Heads-Up Limit Texas Hold'em

Heads-up limit texas hold'em is played according to the following rules:

1. **Blinds:** The game begins with two players posting blinds. The small blind is 5 chips, and the big blind is 10 chips.

2. **Hole Cards:** Each player is dealt two private hole cards.

3. **Deck:** A standard 52-card deck is used, consisting of 4 suits (spades, hearts, clubs, diamonds), each containing 13 cards (2 through Ace).

4. **First Betting Phase (Preflop):** Following the deal of the hole cards, a phase of betting begins with the player to the left of the big blind. The bet size is fixed at 10 chips.

5. **Flop:** After the first betting phase, three community cards (the Flop) are dealt face up in the center of the table.

6. **Second Betting Phase:** A second phase of betting takes place, starting with the player to the left of the dealer. The bet size remains 10 chips.

7. **Turn:** After the second betting phase, a fourth community card (the Turn) is dealt face up.

8. **Third Betting Phase:** A third phase of betting occurs. The bet size increases to 20 chips.

9. **River:** After the third betting phase, a fifth and final community card (the River) is dealt face up.

10. **Fourth Betting Phase:** A final phase of betting takes place. The bet size remains 20 chips.

11. **Showdown:** If no player folds by the end of the final betting phase, both players reveal their hole cards. The player with the highest-ranking hand, using any combination of their hole cards and the community cards, wins the pot. If the hands are tied, the pot is split evenly. Table 4 shows the hand rankings.

12. **Betting Structure:** During each betting phase, players have the option to fold, call, or raise. In each phase, betting is capped at 4 bets (1 bet and 3 raises).

## B  NUMERALL211 HOLD'EM RULES

| Rank | Hand | Prob. (%) | Description | Example |
|------|------|-----------|-------------|---------|
| 1 | Straight flush | 0.321 | 3 of cards with consecutive rank and same suit. Ties are broken by highest card. | T♠9♠8♠ |
| 2 | Three of a kind | 1.587 | 3 of cards with the same rank. Ties are broken by the card's rank. | T♠T♡T♣ |
| 3 | Straight | 4.347 | 3 of cards with consecutive rank. Ties are broken by the highest card rank. | T♠9♡8♣ |
| 4 | Flush | 15.799 | 3 of cards with the same suit. Ties are broken by the highest card rank, then second highest card rank, then third highest card rank. | T♠8♠6♠ |
| 5 | Pair | 34.065 | 2 of cards with the same rank. Ties are broken by the rank of the pair, then by the rank of the third card. | T♠T♡8♣ |
| 6 | High card | 43.881 | None of the above. Ties are broken by comparing the highest ranked card, then the second highest ranked card, and then the third highest ranked card | T♠8♡6♣ |

Table 5: Hand ranks of Numeral211 Hold'em

Numeral211 Hold'em is played according to the following rule:

1. **Ante:** Each player antes 5 chip into the pot at the start of the hand.

2. **Hole Card:** Both players are dealt one private card face down, known as the hole card.

3. **Deck:** The deck consists of a standard poker deck, excluding the Jokers, Kings, Queens, and Jacks, resulting in a total of 40 cards. There are four suits: spades (♠), hearts (♡), clubs (♣), and diamonds (♢), each containing ten cards numbered 2 through 9, and including the ten (T) and ace (A).

4. **First Betting Phase:** Following the deal of hole cards, a phase of betting occurs. Players can choose to check or bet, with the bet size set at 10 chips.

5. **Flop:** After the initial betting phase, a single community card, termed the Flop, is revealed from the deck.

6. **Second Betting Phase:** Another phase of betting takes place after the Flop, with the bet size increasing to 20 chips.

7. **Turn:** After the Second betting phase, another community card, termed the Turn, is revealed from the deck.

8. **Third Betting Phase:** Another phase of betting takes place after the Turn, with the bet size still set at 20 chips.

9. **Showdown:** If neither player folds, a showdown occurs. Players reveal their cards, aiming to form the best possible hand. The player with the highest-ranked hand wins the pot. In the case of a tie, the pot is split evenly. Table 5 show the hand ranks of Numeral211 Hold'em.

10. **Betting Options:** Throughout the game, players have options to fold, call, or raise. In each betting phase, the total sum of bets and raises is limited to a maximum of 4, with fixed bet sizes of 10 chips in the first phase and 20 chips in the last two betting phases.

# C  ALGORITHM DETAILS

## C.1  PSUDOCODE FOR ISOMORPHISM CONSTRUCTOR

Algorithm A1 describes the isomorhpism constructor for isomorphism frameworks (POI, KROI, PWI, KRWI).

---

**Algorithm A1** Isomorphism Constructor

---

**Require:**

$Index_i(r, \cdot) : \Psi_i^{(r)} \mapsto \mathbb{N}$. Signal infoset index function for player $i$.

1: **procedure** ISOMORPHISMCONSTRUCTOR($r$, $\Psi_i^{(r)}$, FEATURE($\cdot$))
2:     Initialize $\mathcal{C}_i^{(r)}$ vector as empty.
3:     Initialize $\mathcal{D}_i^{(r)}$ array arbitrarily with length $|\Psi_i^{(r)}|$.
4:     **for** $\psi \in \Psi_i^{(r)}$ **do**
5:         $feature \leftarrow$ FEATURE($\psi$).
6:         Append $feature$ to $\mathcal{C}_i^{(r)}$.
7:     **end for**
8:     Eliminate duplicates from $\mathcal{C}_i^{(r)}$.
9:     Sort the elements of $\mathcal{C}_i^{(r)}$ in lexicographical order.
10:     Construct hash table $\mathcal{CI}_i^{(r)}$ from $\mathcal{C}_i^{(r)}$. Store the index $lexid$ and value $feature$ of $\mathcal{C}_i^{(r)}$ in $\mathcal{CI}_i^{(r)}$ as key-value pairs ($feature, lexid$).
11:     **for** $\psi \in \Psi_i^{(r)}$ **do**
12:         $feature \leftarrow$ FEATURE($\psi$), $idx \leftarrow Index_i(r, \psi)$.
13:         Update $\mathcal{D}_i^{(r)}[idx]$ with $\mathcal{CI}_i^{(r)}[feature]$.
14:     **end for**
15:     **return** ($\mathcal{C}_i^{(r)}, \mathcal{D}_i^{(r)}$).
16: **end procedure**

---

## C.2  POTENTIAL WINRATE ISOMORPHISM

Algorithm A2 describes the computation process for potential winrate isomorphism. This algorithm operates in reverse, starting from the game's final phase $\Gamma$.

---

**Algorithm A3** K-Recall Winrate Isomorphism

---

**Require:**

$Index_i(r, \cdot) : \Psi_i^{(r)} \mapsto \mathbb{N}$. Signal infoset index function for player $i$.

$\mathcal{PD}_i^{(r)} : \mathbb{N} \mapsto \mathbb{N}$. Potential winrate isomorphism map.

1: **procedure** KRECALLWINRATEISOMORPHISM($\Psi_i, k$)
2:     **for** $r = 1$ to $\Gamma$ **do**
3:         $k' \leftarrow$ MIN($r - 1, k$).
4:         FEATUREFUNC $\leftarrow$ KRECALLWINRATEFEATURE($\cdot, r, k'$).
5:         $(\mathcal{RC}_i^{(r,k')}, \mathcal{RD}_i^{(r,k')}) \leftarrow$ ISOMORPHISMCONSTRUCTOR($r, \Psi_i^{(r)}$, FEATUREFUNC).
6:     **end for**
7:     **return** $(\mathcal{RC}_i^{(1,0)}, \mathcal{RD}_i^{(1,0)}), \ldots, (\mathcal{RC}_i^{(k+1,k)}, \mathcal{RD}_i^{(k+1,k)}), \ldots, (\mathcal{RC}_i^{(\Gamma,k)}, \mathcal{RD}_i^{(\Gamma,k)})$.
8: **end procedure**
9: **procedure** KRECALLWINRATESFEATURE($\psi, r, k$)
10:     initial a empty vector $feature$.
11:     **for** $s = r$ to $r - k$ **do**
12:         $\psi' \leftarrow$ the predecessor signal infoset of $\psi$ in the $s$ phase for player $i$.
13:         $idx \leftarrow Index_i(s, \psi'), abs \leftarrow \mathcal{PD}_i^{(s)}[idx]$.
14:         Append $feature$ with $abs$.
15:     **end for**
16:     **return** $feature$
17: **end procedure**

---

**Algorithm A2** Potential Winrate Isomorphism

---

**Require:**

$Index_i(r, \cdot) : \Psi_i^{(r)} \mapsto \mathbb{N}$. Signal infoset index function for player $i$.

1: **procedure** POTENTIALWINRATEISOMORPHISM($\Psi_i$)
2:     **for** $r = \Gamma$ to $1$ **do**
3:         **if** $r == \Gamma$ **then**
4:             FEATUREFUNC $\leftarrow$ POTENTIALWINRATEFEATURELASTPHASE($\cdot$).
5:         **else**
6:             FEATUREFUNC $\leftarrow$ POTENTIALWINRATEFEATURE($\cdot, r, \mathcal{PC}_i^{(r+1)}, \mathcal{PD}_i^{(r+1)}$).
7:         **end if**
8:         $(\mathcal{PC}_i^{(r)}, \mathcal{PD}_i^{(r)}) \leftarrow$ ISOMORPHISMCONSTRUCTOR($r, \Theta_i^{(r)}$, FEATUREFUNC).
9:     **end for**
10:     **return** $(\mathcal{PC}_i^{(1)}, \mathcal{PD}_i^{(1)}), \ldots, (\mathcal{PC}_i^{(\Gamma)}, \mathcal{PD}_i^{(\Gamma)})$.
11: **end procedure**
12: **procedure** POTENTIALWINRATESFEATURELASTPHASE($\psi$)
13:     **return** $pf_i^{(\Gamma)}(\psi)$         ▷ compute according Equation equation 1
14: **end procedure**
15: **procedure** POTENTIALWINRATEFEATURE($\psi, r, \mathcal{PC}_i^{(r+1)}, \mathcal{PD}_i^{(r+1)}$)
16:     $feature_\psi \leftarrow$ zero array with length $N + 1$
17:     **for** $\psi' \in \Psi_i^{(r+1)}$, such that $\exists \theta' \in \psi', \exists \theta \in \psi: \varsigma(\theta, \theta') > 0$ **do**
18:         $idx \leftarrow Index_i(r + 1, \vartheta'), abs \leftarrow \mathcal{PD}_i^{(r+1)}[idx], feature_{\psi'} \leftarrow \mathcal{PC}_i^{(r+1)}[abs]$.
19:         **for** $j = 0$ to $N$ **do**
20:             $feature_\psi[j] \leftarrow feature_\psi[j] + feature_{\psi'}[j]P(\psi|\psi')$     ▷ Equation equation 2
21:         **end for**
22:     **end for**
23: **end procedure**

---

## C.3   K-RECALL WINRATE ISOMORPHISM

Algorithm A3 constructs the k-recall winrate isomorphism using the k-recall winrate feature. This process requires the prior construction of the potential winrate isomorphism map $\mathcal{PD}_i^{(r)}$ using Algorithm A2.

# D ACCELERATING DISTANCE COMPUTING FOR K-RECALL WINRATE FEATURES

---

**Algorithm A4** Distance Batch

---

**Require:**

$\mathcal{RC}_i^{(r,k)} : \mathbb{N} \mapsto \mathbb{N}^{k+1}$. K-recall winrate feature set.

$\mathcal{PC}_i^{(r)} : \mathbb{N} \mapsto [0,1]^{N+1}$. Potential winrate feature set.

$\mathcal{PD}_i^{(r)} : \mathbb{N} \mapsto \mathbb{N}$. Potential winrate isomorphism map.

$rc = (pc^{(r)}, \ldots, pc^{(r-k)})$. K-recall winrate feature of the input centroid.

**Ensure:**

Distances of all k-recall winrate feature with centroid.

1: **procedure** DISTANCEBATCH($w_0, \ldots, w_k, rc, r, k$)

Initial phase $s$ empty earth mover's distance vector $EmdDis^{(s)}$ for $s = r, \ldots, r - k$.

Initial empty output distance vector $Dis$.

2:     **for** $t = 0$ to $k$ **do**

3:         **for** $pf$ in $\mathcal{PC}_i^{(s)}$ **do**

4:             Append $EmdDis^{(r-t)}$ with Emd($pf, rc[t]$)

5:         **end for**

6:     **end for**

7:     **for** $rfi$ in $\mathcal{RC}_i^{(r,k)}$ **do**

8:         $dis \leftarrow 0$.

9:         **for** $t = 0$ to $k$ **do**

10:             $dis \leftarrow dis + w_t * EmdDis^{(r-t)}[\mathcal{PD}_i^{(r-t)}[rfi[t]]]$.

11:         **end for**

12:         Append $Dis$ with $dis$.

13:     **end for**

    **return** $Dis$.

14: **end procedure**

---

KrwEmd is based on the KMeans++ clustering algorithm, where in each iteration, the distance (equation 5) between every centroid and each k-recall winrate feature must be calculated. The centroids are predefined, but the scale of k-recall winrate features varies depending on the game. For instance , as shown in Table 2, in the River phase of HULHE, this number reaches an astounding 577366243. Computing the distance involves performing k Earth Mover's Distance (EMD) calculations for every centroid-feature pair, which is highly computationally expensive.

It's important to note that k-recall winrate features are actually combinations of multiple potential winrate features. To optimize the process, we first calculate the EMD between centroids and potential winrate features in the corresponding phase. We then express the distance between centroids and k-recall winrate features as a linear combination of these precomputed EMDs.

Algorithm A4 is responsible for computing the distance between a given centroid $rc = (pc^{(r)}, \ldots, pc^{(r-k)})$ and all k-recall winrate features, where $rc[t] = pc^{(t)}$ represents the potential winrate feature in phase $t$. Lines 2-5 enumerate all potential winrate features in phase $t$ for the centroid and compute the corresponding EMD distance. Lines 7-12 indicate that, for a k-recall winrate feature, it is sufficient to retrieve its corresponding k+1 potential winrate features and, using precomputed distances, apply the weights $w_0, \ldots, w_k$ to obtain the centroid's distance to that k-recall winrate feature. This approach reduces the computational burden of EMD to the scale of potential winrate features. For example, in the River phase of HULHE and HUNL, this optimization results in a compression ratio of $\frac{169 + 1028325 + 1850624 + 20687}{577366243} = 0.0050225$, substantially reducing the computational cost.

However, it must be acknowledged that the overall complexity of the KrwEmd distance calculation still depends on the scale of the k-recall winrate features, as determined by lines 7-13, which remains a significant computational expense.

# E  SUPPLEMENTARY DATA FOR ISOMORPHISM FRAMEWORKS EXPERIMENT

Figure 4 show all of the result in isomorphism frameworks experiment.

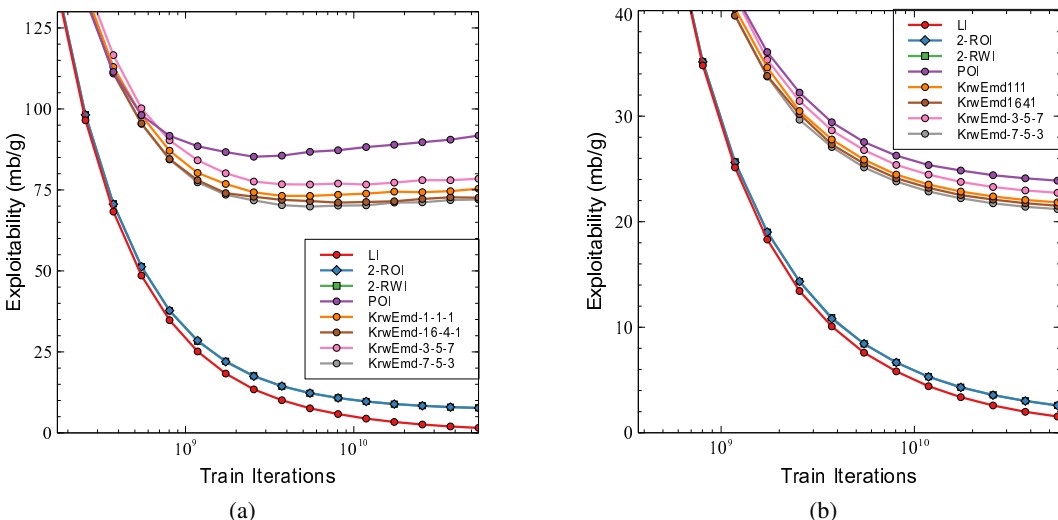

Figure 4: All data within experiment 1

