# OpenReview forum: "KrwEmd: Revising the Imperfect Recall Abstraction from Forgetting Everything"
_ICLR.cc/2025/Conference — Submitted to ICLR 2025_

### Official Review · Reviewer_7mWG · 2024-10-24

**Soundness:** 3
**Presentation:** 2
**Contribution:** 2
**Rating:** 6
**Confidence:** 2

**Summary:**

This paper introduces KrwEmd, a novel algorithm designed to address the excessive abstraction issue in solving games with ordered signals, a subset of imperfect information games. This issue arises from extreme implementations of imperfect recall, which discard historical information. To this end, the authors introduce the k-recall winrate feature, which combines future and historical game information to better distinguish between different game states. Then they propose the KrwEmd algorithm, which uses Earth Mover’s Distance (EMD) to cluster game states more effectively by considering winrate feature. Finally, experimental results show that KrwEmd outperforms other algorithms, particularly in maintaining game information and delivering better decisions.

**Strengths:**

1. The KrwEmd algorithm is a novel method for integrating historical information into game abstraction through the k-recall winrate feature. It addresses the significant issue of excessive abstraction in games with ordered signals. This creative combination of historical data with future signal-based clustering represents a significant improvement over traditional methods that rely solely on future information.

2. The paper demonstrates high-quality execution, with solid experimental validation.  The comparisons to state-of-the-art methods, such as POI and other imperfect recall algorithms, are thorough, and the results consistently show the advantages of KrwEmd.  The well-structured experiments provide convincing evidence that the algorithm improves performance while maintaining a reasonable computational cost.

3. The paper makes a meaningful contribution to the field of imperfect information games, as it solves a long-standing issue of excessive abstraction, which is critical for improving AI performance in games like poker and other strategic scenarios.

**Weaknesses:**

1.	The organization of the paper could be improved for better clarity. For instance, the relationship between Section 4 and Section 5 is not clearly explained, making it difficult for readers to follow the logical progression of the content.

2.	The introduction of the KrwEmd algorithm is not very clear. Providing a step-by-step breakdown of the algorithm with examples or diagrams could make the technical details easier to understand. Important algorithms, such as those essential to understanding the core contribution, are relegated to the Appendix, which diminishes their visibility.

**Questions:**

As I’m not an expert in this area, I have listed my questions below. If there are any mistakes, please feel free to correct me.

Q1: Given that the algorithm is tested only on the Numeral211 Hold’em testbed, do you have plans to test it on a broader range of games, or whether this algorithm can be applied to other games? If so, what kinds of games or environments would you target next, and why were they not included in the current experiments?

Q2: Given that the k-recall feature is manually tuned, have you considered an adaptive or automated mechanism for hyperparameter tuning?

Q3: The paper introduces historical information, but there isn’t much discussion on the trade-off between adding more history and the increase in complexity. Could you clarify how you determined the optimal amount of historical recall to use? In other words, how does the choice of k affect the algorithm’s performance across different games?

Q4: In games with deeper decision trees, especially in the late stages, how does the historical information influence decision-making? Is there a risk that using too much historical information could overcomplicate the abstraction process in the later stages of the game?

---

### Official Review · Reviewer_PmhZ · 2024-11-04

**Soundness:** 1
**Presentation:** 1
**Contribution:** 2
**Rating:** 3
**Confidence:** 2

**Summary:**

State abstractions allow one to plan in complex sequential decision-making settings by grouping states so as to reduce the computational requirements of search. However, in certain cases in imperfect information, improper abstractions may lead to a reduction in performance. This paper introduces an approach to address these extreme cases. The paper introduces a new feature (k-recall winrate) for signal infostates that takes into account both future data and histories and provides a measure of similarity between infosets. The work combines this feature with the earth mover's distance as a distance measure between infosets and shows that this helps improve agent performance.

**Strengths:**

**Originality:** The paper introduces a novel infoset feature and combines with an existing measure to perform state abstractions.

**Significance:** I cannot gauge the impact of the paper (explained shortly).

**Quality:** The experimental set up appears reasonable. The authors compared their method against other baselines (see comment in weaknesses) relative to a common adversary.

**Weaknesses:**

The major problem with this paper is that it is prohibitively difficult to read. Perhaps I may not be the exact target reader, but I found it impossible to read Sec 2.1. It was a wall of notation and definition of which I was unable to keep track. It is therefore difficult for me to comment on the soundness of the algorithm proposed. However, I continued to try to complete the other parts of the paper with the idea of the algorithm I gathered from what I could understand.

Pertinent details about the experiments do not seem to be provided (games per training iteration or what they meaning by a training iteration, the number of seeds over which the train, significance testing.) Absent this information, I cannot say whether the results support the authors' claims they outperform the other algorithms.

I do not know if it is possible to write the paper as such, but it would be easier to understand if the notations were introduced gradually instead of all at once. For example, perhaps discussing notation needed to understand games then move on to introducing signals. Building up the background in parts might go a long way in easing the average reader into understanding what is needed.

**Questions:**

I do not think there is anything I could ask that would help me understand the paper.

---

### Official Review · Reviewer_GjCX · 2024-11-05

**Soundness:** 2
**Presentation:** 2
**Contribution:** 1
**Rating:** 3
**Confidence:** 3

**Summary:**

In this paper, authors proposed a solution for a hand abstraction task in a simplified Texas Holdem game. Their work is based on arXiv paper by Fu et al. (2024) which introduced the use of historical information. This paper highlights the use of practical signal abstraction algorithms.

**Strengths:**

* They revisit the research topics studied in the early 2000s on hand abstraction in poker games.
* They proposed to use win-rate isomorphism for the imperfect recall abstraction.
* They tested their algorithm with the variants of hold'em game and compared their work with other works.
* They provided algorithm details in the appendix section.

**Weaknesses:**

The major concern of this paper is relationship with the arXiv paper by Fu et al. (2024). It looks like this paper extends the Fu et al. (2024) arXiv paper and there are some overlaps on ideas, formulation, description and experimental setups.

Because Fu's work is not yet published in a peer-reviewed conference or journal (at this time), it's not yet fully evaluated by the academic community. It makes this work needs more explanation on the soundness of their approach.

**Questions:**

* What's the relationship with your work and Fu et al. (2024) work? Is your work significantly enhanced by the arXiv paper?
* Why is your work the first algorithm in the context of the practical side?
* Please explain about the potential applications or implications of your work beyond poker games, such as how your approach to incorporating historical information in abstraction tasks might be relevant to other areas of machine learning or AI that ICLR researchers work on.
* Please discuss the potential challenges and adaptations needed to apply your algorithm to other domains or game types. Additionally, please explain why you chose this specific testbed and how it relates to more complex real-world scenarios.

---

### Meta-Review · Area_Chair_fCaR · 2024-12-23

**Metareview:**

The work presented in this paper may or may not be worth publishing, but the current paper is very hard to follow, and may be derivative of other work, in particular a recent paper by Fu et al. The authors do not address the potential overlap here, and do in fact not engage with the reviewers at all. Given the lack of engagement, I see no choice but to reject.

**Additional Comments On Reviewer Discussion:**

The authors did not engage with the reviewers at all, so I must presume they are not interested in getting their paper accepted.

---

### Decision · Program_Chairs · 2025-01-22

Reject